# Hybrid Intelligence for Marine Biodiversity: Integrating Citizen Science with AI for Enhanced Intertidal Conservation Efforts at Cape Santiago, Taiwan

**Vincent Y. Chen** [1,*], **Day-Jye Lu** [2] and **Yu-San Han** [1,*]

1   Institute of Fisheries Science, College of Life Science, National Taiwan University, No. 1, Sec. 4, Roosevelt Rd., Taipei 106216, Taiwan
2   School of Forestry & Resource Conservation, National Taiwan University, Taipei 106319, Taiwan; djlu@ntu.edu.tw
*   Correspondence: d10b45001@ntu.edu.tw (V.Y.C.); yshan@ntu.edu.tw (Y.-S.H.); Tel.: +886-3-33663726 (V.Y.C.); +886-2-33663726 (Y.-S.H.); Fax: +886-3-33669449 (V.Y.C.); +886-2-33669449 (Y.-S.H.)

**Abstract:** Marine biodiversity underpins the formation of marine protected areas (MPAs), necessitating detailed surveys to account for the dynamic temporal and spatial distribution of species influenced by tidal patterns and microhabitats. The reef rock intertidal zones adjacent to urban centers, such as Taiwan's Cape Santiago, exhibit significant biodiversity, yet they are increasingly threatened by tourism-related activities. This study introduces an artificial intelligence (AI)-empowered citizen science (CS) approach within the local community to address these challenges. By integrating CS with AI, we establish a hybrid intelligence (HI) system that conducts in situ biological surveys and educational programs focused on reef ecological conservation. This initiative not only facilitates the collective gathering and AI-assisted analysis of critical data but also uses machine-learning outputs to gauge data quality, thus informing subsequent data collection and refinement strategies. The resulting collectivity and iterative enhancement foster a mutual and continuous HI learning environment. Our HI model proves instrumental in fostering community engagement and public involvement in CS endeavors, cultivating the skills necessary for documenting rocky intertidal biodiversity shifts. These efforts are pivotal for informing the design and governance of future MPAs, ensuring their efficacy and sustainability in marine conservation.

**Keywords:** marine protect areas; rocky intertidal ecosystem; hybrid intelligence; citizen science; artificial intelligence

## 1. Introduction

Coastal regions, including the marine littoral zone, are biological powerhouses [1]. They offer refuge to an expansive array of marine life and directly benefit humanity in multifarious ways [2]. From food sources and economic avenues to recreational pursuits and the conservation of cultural heritages, these ecosystems are indispensable [3]. In parallel, rocky intertidal systems, especially those adjacent to urban locales, are paramount to sustaining ecological balances while also being focal points for economic, social, and recreational engagements [4].

However, the allure of these regions is both their strength and vulnerability. The recent upsurge in coastal tourism and its resultant economic windfall for local regions [5,6] draw legions of tourists. Urban inhabitants, especially, are enchanted by the serene coastal landscapes [7] and the prospect of close encounters with marine fauna. Yet, this very attraction coupled with activities like marine organism harvesting and heedless trampling cast shadows of environmental stress on these habitats [8–10]. The rich biodiversity, a hallmark of these zones, faces relentless challenges. Rocky intertidal areas, rife with microhabitats, stand as evidence of this. The seemingly benign acts of visitors, such as

walking over tide pools or curiously overturning rocks, inflict lasting ecological scars. The cascading effects of such disturbances manifest in dwindling algal populations, impacting both fleshy and coralline types [11], and in the perceptible strain on marine entities, like echinoderms [12], mollusks [13,14], and crustaceans [15].

The Anthropocene epoch, marked by significant human impact on the Earth's biophysical systems, has led to intensified pressures on coastal regions due to population growth, urbanization, and climate change [16]. This has notably affected rocky intertidal ecosystems with human interaction evolving from fishery exploitation to recreational activities like adventure tourism [17]. A multifaceted coastal management approach, incorporating diverse strategies from legal to educational, is essential to mitigate environmental impacts and encourage societal participation [18]. Additionally, the largely unexplored marine domain presents opportunities for marine citizen science (MCS). MCS plays a crucial role in filling research gaps and enhancing global marine conservation efforts [19,20]. It provides cost-effective, robust data that inform policy decisions [21,22] while increasing public science literacy and community engagement in marine issues [23–25] from cetacean conservation [26,27] to addressing plastic pollution [28], thereby enabling community-driven initiatives to transform research into effective policies [29,30].

At the heart of this movement is the citizen science (CS) research methodology that enlists the general public in data collection, categorization, or scientific analysis [31]. Historically, it has been instrumental in various marine research undertakings. Notable projects include monitoring reef fish ecosystems [32], tracking queen conch species populations [33], discerning seagrass bed dynamics [34], and mapping the distribution patterns of marine litter [35,36]. The unique advantage of CS in marine projects lies in its capacity to amplify the spatial and temporal scope of studies [37,38].

Yet, the integration of projects/research with CS is not without challenges. Concerns often center on the data quality when gathered by non-professionals, especially for intricate biodiversity datasets, which could potentially impede their application [39]. Nevertheless, recent innovations offer solutions. As highlighted by Earp et al. [40], implementing stringent protocols, ensuring thorough training, and adhering to meticulous data verification processes can elevate the data's quality. With such measures in place, data sourced from citizen scientists can rival if not match those procured by their professional counterparts, as evidenced in their intertidal algae ecology experiment.

AI has been employed to augment and enhance human understanding of the environment, including perceptions of citizen scientists [41]. While CS and AI are often viewed as separate tools for ecological monitoring, recent studies indicate that a symbiotic relationship between human intelligence and AI, termed hybrid intelligence (HI) [42,43], can strategically unite the two, enhancing outcomes for conservation activities. By pairing the public engagement benefits of CS projects with the sophisticated analytical prowess of AI, there is the potential to foster greater multi-stakeholder consensus on matters of public [44] and scientific importance. Moreover, the integration of both methodologies can expedite data collection and processing relative to traditional scientific approaches, indicating a promising avenue for accelerated monitoring and conservation efforts [45].

Cape Santiago, located on Taiwan Island's easternmost point near Taipei, is at the intersection of the Kuroshio and longshore currents. Named by Spanish explorers 400 years ago, its 4.86 km coastline features wave-cut benches and rocky shores and is home to the small Magang fishery harbor community. With declining fishery resources driving youth to the cities, the local government, aiming to boost tourism, built a bicycle route to the coast in 2011. However, the influx of tourists engaging in harmful activities like capturing marine specimens has raised concerns about the impact on biodiversity and the coastal environment. The Cape Santiago Culture Development Association (SDCDA) was formed to protect the fishery village culture and marine ecosystem. Legally, Cape Santiago's coast is an "Ocean Resource Protected Area" under the Urban Planning Law [46], restricting construction but not specifically addressing environmental harm.

Acknowledging the existing shortcomings in marine ecological conservation, the Ocean Affairs Council introduced the Marine Conservation Act proposal in 2019 [47]. Draft article 7 of this act empowers management authorities to designate coastal regions with significant biodiversity that require special protection as marine sanctuaries. They are also mandated to implement management measures such as ecological monitoring, as detailed in draft article 6. Thus, the SDCDA is advocating for the wave-cut bench on the western side of the fishery harbor to be recognized as a strictly regulated marine protected area. This designation would facilitate the management of environmentally detrimental actions under the Marine Conservation Act. The scientific assessment of biodiversity along this coast will be a crucial factor in determining if the area qualifies as a marine protected zone. The community members are unified in their stance and are mobilizing volunteers for coastal biodiversity CS surveys. Before furnishing the necessary scientific data to legislative bodies for the inclusion of this area as a marine protected zone, it is imperative to proactively educate tourists on the coastal ecosystem. This initiative aims to teach visitors the importance of marine conservation, discouraging activities that can harm marine life and the environment.

This study aims to assist the SDCDA in employing a CS project and AI for the conservation of Cape Santiago's intertidal biodiversity. This study has set the following objectives:

1. Create a spatiotemporal biodiversity database for the benches of Cape Santiago;
2. Assess the performance of the AI model, the data contributions from CS, and the overall impact of the collaboration;
3. Formulate an HI framework that merges CS and AI, targeting continuous monitoring and environmental education.

## 2. Methods

### 2.1. Study Area and the Citizen Scientist Project

The study area of this paper is located on the northwest side of the Cape Santiago coastline, specifically focusing on a wave-cut bench, termed the Santiago bench. This bench spans approximately 160 m in length and 70 m in width (Figure 1). The primary aim of the survey is to identify and document target marine organisms within this predefined area.

To evaluate the potential marine protected areas (MPAs), the Ocean Consecration Administration (OCA) conducted biodiversity assessments at 67 rocky shores spanning Taiwan's main island and its surrounding islands [48]. They utilized transect line and quadrat methodologies to assess marine species richness, biomass, and a range of environmental factors, including the impacts of sewage outfalls, man-made facilities, tourist activity, siltation levels, and the abundance of rock pools. By integrating both biodiversity and environmental indicators, the OCA determined the conservation priorities of these 67 rocky shores. Among the surveyed locations were four sites at Cape Santiago, including Santiago bench, Lai-Lai-1, Lai-Lai-2, and Lai-Lai-3. Notably, the Lai-Lai-2 site, located at the southeastern edge of Cape Santiago, showcased the greatest marine species diversity. Along with the neighboring Lai-Lai-3 site, both were identified as having the highest conservation value, as reported in the OCA's pilot survey [48].

The participants in the CS project were recruited by the SDCDA and underwent a comprehensive 16 h training session. This training workshop, conducted by the authors in collaboration with marine experts, encompassed topics such as rocky shore ecology, macrobenthos species identification (Table 1), AI training procedures, survey assistance methods, and environmental education. To enhance their training, materials from a prior survey compiled from the author's dataset [49] that identified 13 phyla and cataloged 234 marine species, including a new marine flatworm species [50], were provided, complemented by images of each species previously documented in the region. The main responsibilities assigned to the CS participants were twofold: data collection through field surveys and public education regarding the intertidal environment.

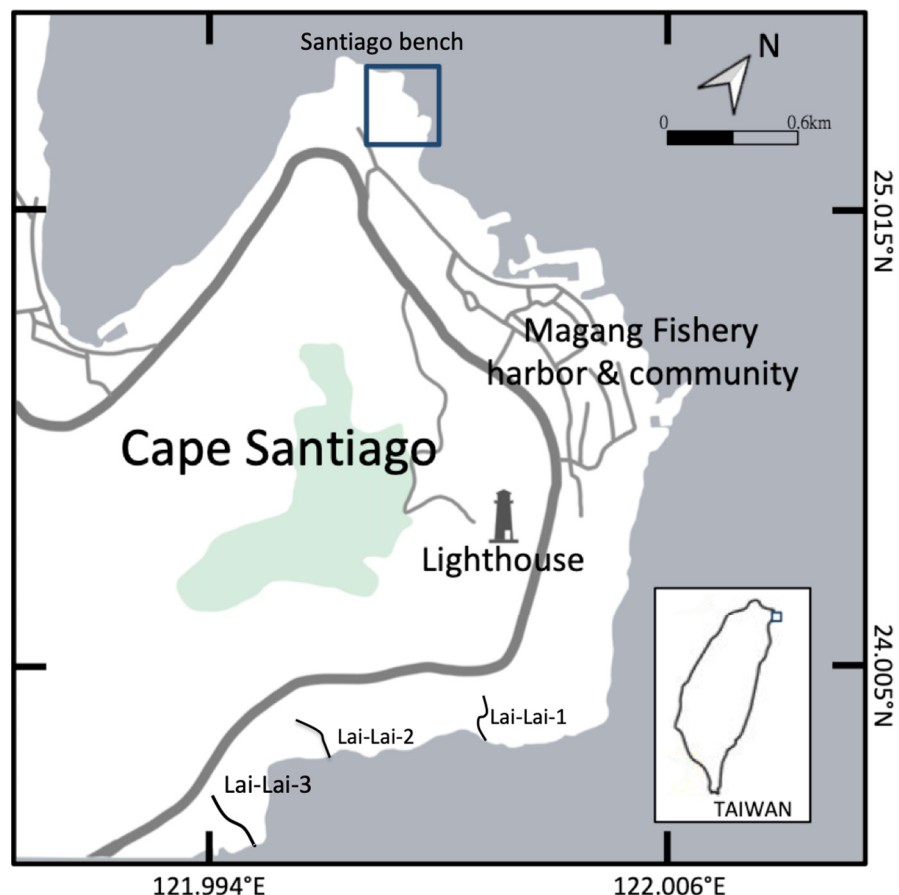

**Figure 1.** Location of the study site at Cape Santiago with its position within Taiwan highlighted (inset) and the three survey lines conducted by the OCA.

**Table 1.** The structure of the macrobenthos in the study area of marine species of Santiago bench.

| Kingdom | Phylum | Common Names | Species Numbers |
|---|---|---|---|
| Animalia | Porifera | Sponges | 8 |
| | Bryozoa | Bryozoans, Moss Animals | 4 |
| | Mollusca | Cowries, Chitons, Sea Slugs, Octopuses, Oysters | 72 |
| | Arthropoda | Crabs, Shrimps, Barnacles | 22 |
| | Annelida | Segmented Worms | 8 |
| | Coelenterate | Corals, Sea Anemones, Jellyfishes | 10 |
| | Echinoderms | Sea Stars, Sea Urchins, Sea Cucumbers | 27 |
| | Nemertina | Ribbon Worms | 2 |
| | Platyhelminthes | Flatworms | 8 |
| | Chordata | Tunicates, Vertebrates, Fishes, Sea Snakes | 39 |
| Plantae | Chlorophyta | Green Algae | 9 |
| | Rhodophyta | Red Algae | 19 |
| | Chromophyta | Brown Algae | 6 |

The survey protocol, derived from the Ocean Consecration Administration's (OCA) transect line method, required two individuals to methodically search and document marine species along a designated transect line, spanning from the high tide zone to the

low tide area of the shore. The survey procedure was conducted by volunteers who had previously received a CS training workshop. During the survey, the participants followed a pre-set route marked by the transect line, photographing all marine specimens encountered and recording the quantities of specific target species on survey forms. Following the survey, the collected data and images were systematically compiled and uploaded to create a comprehensive training dataset for future artificial intelligence (AI) training sessions.

*2.2. Training AI Model and AI Performance Evaluations*

The convolution neural network (CNN) is a cornerstone in AI models for image pattern extraction and recognition [51]. R-CNN and YOLO, both built on the CNN framework, can distinguish various objects within an image [52,53]. The YOLOv5s [54] model was chosen for this study because of its high calculating speed and accuracy in detecting marine life in natural settings. YOLOv5s, being a deep-learning approach in machine vision, requires comprehensive training on an extensive image dataset to perform optimally [55,56]. The images of organisms from the participants' field survey dataset were meticulously identified and annotated as 'instances' using the open-source Python program labelImage (https://github.com/HumanSignal/labelImg, accessed on 10 July 2022). Each instance was marked as a bounding box with its location, size, and common name within the image. Effective AI recognition of a particular specimen necessitates a substantial image database for that specimen, typically comprising 200 or more instances, as per our experience. Consequently, species that are commonly observed at the research site are usually selected for AI training. To keep participants engaged, we also include a selection of rare target species, like sea hares and blue-ringed octopuses, encouraging participants to search for and document these less frequently encountered organisms. These annotated images were then compiled into a training dataset for YOLOv5s. All datasets were randomly split with 80% allocated for training and 20% for validation. Training was conducted on Linux desktop computers equipped with an RTX3090-24G GPU, using hyperparameters that encompassed a pre-trained weight, a training batch size of 32, an image resolution of $800 \times 800$, and a learning epoch of 200.

Two primary tasks need to be executed by a trained AI model. The first involves identifying target species instances from an image, while the second requires the accurate prediction of the specific species' common name, distinguishing it from other species. The AI model's predictions for any given specimen can be categorized into four possible outcomes:

4. Correctly identifying the actual species—true positive (TP).
5. Accurately recognizing a species other than the target—true negative (TN).
6. Missing the identification of the actual species—false negative (FN).
7. Incorrectly labeling a different species as the target species—false positive (FP).

The outcomes of distinguishing between various species can be depicted in a two-dimensional table, commonly referred to as a confusion matrix. In this matrix, the actual species are represented on the *x*-axis, while the predicted species appear on the *y*-axis (refer to Figure 2). The overall model's performance was assessed using Equations (1), (2), and (4):

$$precision = \frac{TP}{TP + FP} \tag{1}$$

$$recall = \frac{TP}{TP + FN} \tag{2}$$

To calculate the ratio of the area in the predicted instance bounding box $B_p$ overlapping the area of the actual ground truth box $B_g$, the intersection of union, *IoU*, formula is used as follows:

$$IOU = \frac{aera \ |Bp \cap Bgt|}{aera \ |Bp \cup Bgt|} \tag{3}$$

The mAP$_{0.5}$ Equation (3) measures the averaged precision value of all the cases when the *IOU* is 50% threshold area overlap.

$$mAP_{0.5} = \frac{1}{n}\sum_{i=1}^{n} AP_{category\ i} \quad when\ IoU\_threshold = 0.5 \tag{4}$$

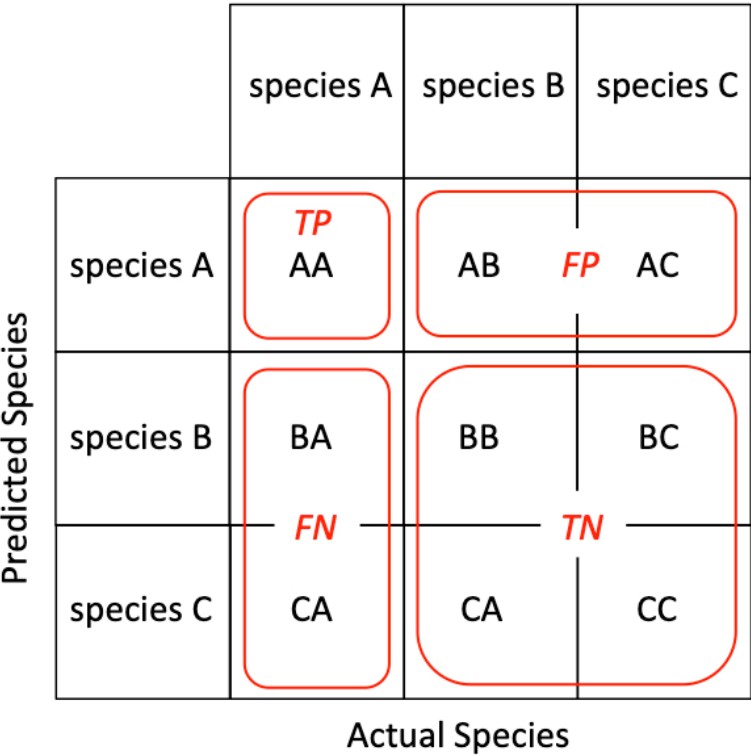

**Figure 2.** An example of a 3 × 3 multi-class confusion matrix with actual species as *x*-axis/row grids and predicted species as *y*-axis/column grids. While focusing on class A, the AA grid is true positive, TP, the rest of the row grids are false positive, FP, the rest of the column grids are false negative, FN, and the rest of the grids except all above grids are all true negative, TN.

### 2.3. Hybrid Intelligence Criteria for CS and AI Synergy

To investigate the potential synergy between CS and AI in forming HI, we adopted the operational definition of hybrid intelligence (HI) from [57], which is based on three criteria:

8. **Collectiveness:** This emphasizes the collaboration between humans and AI with the aim of collectively addressing a task to achieve a system-level goal. It is acknowledged that individual agents might have sub-goals that deviate from the overarching system objective.
9. **Solution Superiority:** This asserts that the combined sociotechnical system, which includes both humans and AI, produces results surpassing what individual agents, whether human or AI, could achieve independently.
10. **Mutual and Continuous Learning:** The system shows consistent improvement, both collectively and at the individual component level (human and AI). This continuous enhancement signifies persistent learning and development from both parties.

## 3. Results

### 3.1. Survey Protocols

Initially, the survey protocol was adapted from the OCA's methodology [48]. This approach involved setting up 10 transect lines on the Santiago bench within the determined survey zone. The survey groups, each comprising two members, aligned with these lines. Therefore, a total of 10 groups worked in tandem, collecting data and photographs for AI

training. The primary objective was to document the time and spatial occurrence of marine species. Upon implementation, some challenges with the original proposal survey protocol (PSP) (Figure 3a) became apparent. Some species, like the sea hare, were absent during the summer surveys. Considering the route's dual function as a prospective educational pathway, a looped configuration was determined to be more suitable than a linear design. This feedback led to the development of an adaptive survey protocol (ASP). The pivotal change in the adaptive survey protocol involved the survey route. Instead of traversing directly from the shore to the sea, the route now followed grooves (GRV) parallel to the coastline. These grooves acted as paths that led to four tidal pools (TP) and four grooves (GRV). This revised protocol offered flexibility in group sizes, and the teams could now conduct their surveys asynchronously, adhering to the predetermined route (Figure 3b).

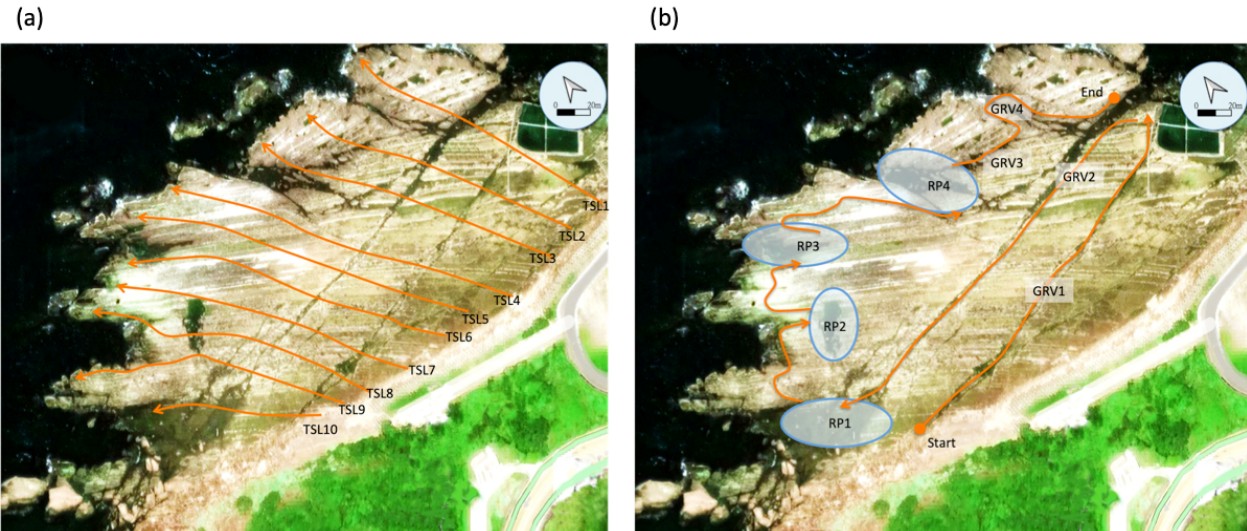

**Figure 3.** The survey transect routes of different survey protocols. (**a**) The routes of ten transect lines (TSL) of the proposed survey protocol (PSP), and (**b**) a loop comprised of 4 grooves (GRV) and 4 rock pools (RP) of the adaptive survey protocol (ASP).

The survey data are categorized based on both temporal and spatial factors. Temporal data capture the season during which the survey was conducted, categorized into spring (SP), summer (SU), autumn (F), and winter (W). The surveys were uniquely timed to coincide with low tide moments, allowing for the CS participants to access the exposed bench. Spatial data are differentiated based on tide levels: high tide level (HTL) and low tide level (LTL). As the tide recedes and the Santiago bench is unveiled, areas retaining water in depressions are labeled as LTL, while areas showcasing organisms above the water surface are designated as high water level (HWL). These designations aid in recording the spatial distribution of marine organisms (Figures 4a and 5a).

### 3.2. Proposal Survey Protocol and AI Training

From the proposed survey protocol, we garnered a collection of 1301 photographs that showcased nine distinct marine species, complete with their spatial and temporal distribution. The total annotated instances across all specimens amounted to 2643 (Figure 4a). Utilizing this dataset, we trained the PSP YOLOv5s neural network, achieving a performance marked by a precision of 0.96175, recall of 0.93533, and mAP_0.5 of 0.95485. An examination of the confusion matrix indicated high true positive (TP) values for almost all species. Notably, barnacles and oysters registered TP values of 0.83 and 0.87, respectively. All other species achieved impressive TP values exceeding 0.97 with some even reaching a perfect score of 1.0 (100%) (Figure 4b).

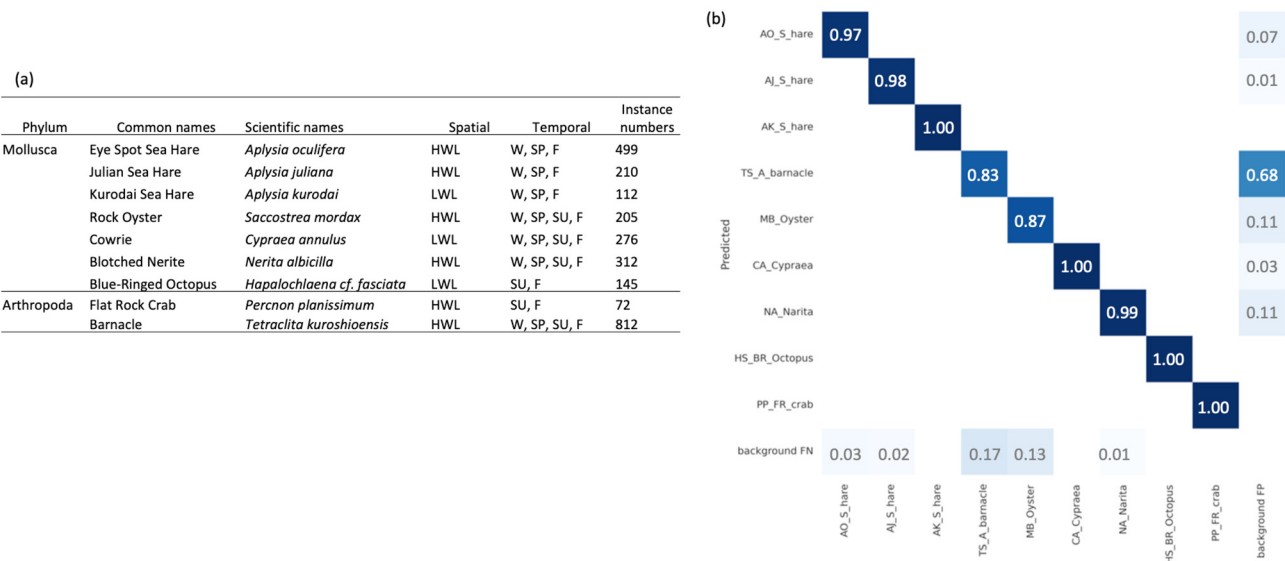

**(a)**

| Phylum | Common names | Scientific names | Spatial | Temporal | Instance numbers |
|---|---|---|---|---|---|
| Mollusca | Eye Spot Sea Hare | *Aplysia oculifera* | HWL | W, SP, F | 499 |
| | Julian Sea Hare | *Aplysia juliana* | HWL | W, SP, F | 210 |
| | Kurodai Sea Hare | *Aplysia kurodai* | LWL | W, SP, F | 112 |
| | Rock Oyster | *Saccostrea mordax* | HWL | W, SP, SU, F | 205 |
| | Cowrie | *Cypraea annulus* | LWL | W, SP, SU, F | 276 |
| | Blotched Nerite | *Nerita albicilla* | HWL | W, SP, SU, F | 312 |
| | Blue-Ringed Octopus | *Hapalochlaena cf. fasciata* | LWL | SU, F | 145 |
| Arthropoda | Flat Rock Crab | *Percnon planissimum* | HWL | SU, F | 72 |
| | Barnacle | *Tetraclita kuroshioensis* | HWL | W, SP, SU, F | 812 |

**Figure 4.** (**a**) Displays the species common name, scientific name, spatial distribution (high water level (HWL) or low water level (LWL)), temporal distribution (spring (SP), summer (SU), autumn (F), winter (W)), and instance number of 9 species collected and used for training YOLOv3 following adaptive protocol surveys conducted by CS participants. (**b**) Shows the confusion matrix true positive (TP) results for different species classes.

**(a)**

| Kingdom | Phylum | Common names | Scientific names | Spatial | Temporal | Instance number |
|---|---|---|---|---|---|---|
| Animalia | Mollusca | Chitons | *Liplophura sp.* | LWL | W, SP, F | 195 |
| | | Pyramid Periwinkle | *Nodilittorina pyramidalis* | HWL | W, SP, F | 737 |
| | | Eye Spot Sea Hare | *Aplysia oculifera* | HWL | W, SP, F | 489 |
| | | Julian Sea Hare | *Aplysia juliana* | HWL | W, SP, F | 210 |
| | | Kurodai Sea Hare | *Aplysia kurodai* | LWL | W, SP, F | 112 |
| | | Rock Oyster | *Saccostrea mordax* | HWL | W, SP, SU, F | 205 |
| | | Cowrie | *Cypraea annulus* | LWL | W, SP, SU, F | 276 |
| | | Blotched Nerite | *Nerita albicilla* | LWL | W, SP, SU, F | 312 |
| | | Blue-Ringed Octopus | *Hapalochlaena lunulata* | LWL | SU, F | 145 |
| | | Potamidids | *Batillaria sp.* | HWL | W, SP, F | 1266 |
| | Arthropoda | Sea Roach | *Ligia taiwanensis* | HWL | SU, F | 478 |
| | | Swimming Crab | *Thalamita prymna* | LWL | W, SP, SU, F | 275 |
| | | Pebble Crab | *Eriphia ferox* | HWL | W, SP, SU, F | 202 |
| | | Xantho Crab | *Xantho sp.* | LWL | W, SP, SU, F | 192 |
| | | Flat Rock Crab | *Percnon planissimum* | HWL | SU, F | 72 |
| | | Barnacle | *Tetraclita sp.* | HWL | W, SP, SU, F | 804 |
| | Echinoderms | Black Sea Cucumber | *Holothuria atra* | LWL | SU, F | 295 |
| | | Brittle Star | *Ophiocoma dentata* | HWL | W, SP, SU, F | 213 |
| | Chordata | Goby | *Bathygobius cocosensis* | LWL | W, SP, SU, F | 632 |
| | | Crescent Grunter | *Terapon jarbua* | LWL | W, SP, SU, F | 298 |
| | | Damsel Fish | *Abudefduf sexfasciatus* | LWL | W, SP, SU, F | 221 |
| | | Tunicates | *Polycitor cf. proliferus* | LWL | W, SP, F | 512 |
| | Porifera | Sponges | *Callysponia sp.* | LWL | W, SP, SU, F | 267 |
| Plantae | Chlorophyta | Sea Lettuce | *Ulva lactuca* | HWL | W, SP, F | 640 |
| | Rhodophyta | Red Coral Algae | *Corallina pilulifera* | LWL | SU, F | 301 |
| | Chromophyta | Sargassum Seaweed | *Sargassum sp.* | LWL | SP, SU, F | 781 |
| | | Ballweed | *Colpomenia sinuosa* | LWL | SP, SU, F | 280 |

**Figure 5.** *Cont.*

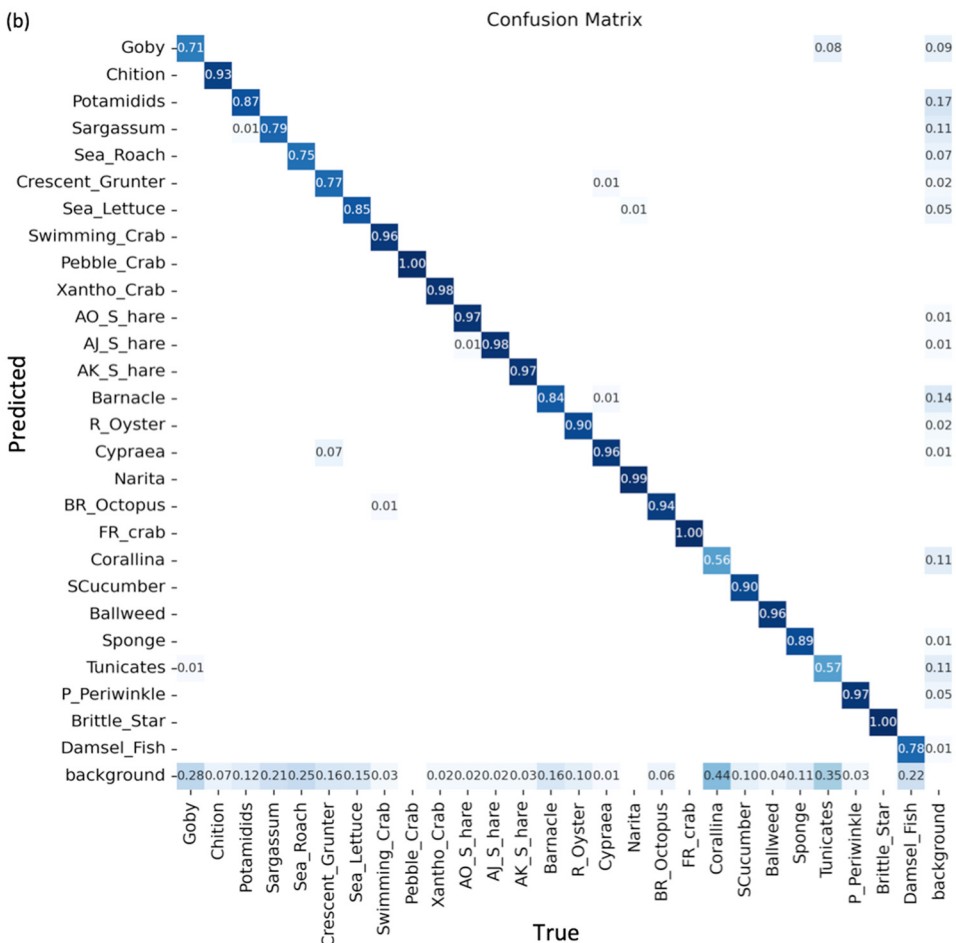

**Figure 5.** (**a**) Displays the species common name, scientific name, spatial distribution (high water level (HWL) or low water level (LWL)), temporal distribution (spring (SP), summer (SU), autumn (F), winter (W)), and instance number of 27 species collected and used for training YOLOv3 following adaptive protocol surveys conducted by CS participants. (**b**) Shows the confusion matrix true positive (TP) results for different species classes. The darker color shows the higher TP accuracy.

### 3.3. Adaptive Survey Protocol and AI Training

Based on the integration of the ASP and the earlier PSP, we obtained a comprehensive collection of 5461 photographs that depict 27 distinct marine species. This dataset not only showcases the spatial and temporal distribution of these species but also includes a total of 7729 annotated instances (Figure 5a). Leveraging this enriched dataset, we trained the ASP YOLOv5s model, which yielded a performance evaluation with a precision of 0.87412, recall of 0.85843, and mAP_0.5: of 0.8707. A deeper dive into the confusion matrix, post enhancement of instance and species counts, revealed that the true positive (TP) values for species like red coral algae (*Corallina* sp.) and tunicates were the lowest, registering at 0.56 and 0.57, respectively. Notably, most of the species TP values from the PSP remained unchanged. The earlier low TP values observed in the PSP dataset for barnacles and oysters showed slight improvements, rising from 0.83 to 0.84 and 0.87 to 0.90, respectively (Figure 5b).

### 3.4. Deployment of AI for In Situ Identification

An application program interface (API) was developed using the Python programming language to facilitate a web-based interface (Figure 6). This interface hosts the trained AI model dedicated to marine specimen recognition and is stationed on a cloud server for efficient specimen identification tasks. Crafted with the intent to aid the CS participants

during surveys or environmental educational endeavors, this tool proves invaluable in real-time specimen identification. On encountering marine organisms in their natural habitat, users can snap photos using their mobile devices and promptly upload them for instant AI-based identification. Upon receiving the image, the API communicates with the AI model to carry out the recognition process, subsequently relaying the results back to the user (Figure 7). This streamlined process greatly enhances the efficiency and accuracy of specimen surveys and educational activities.

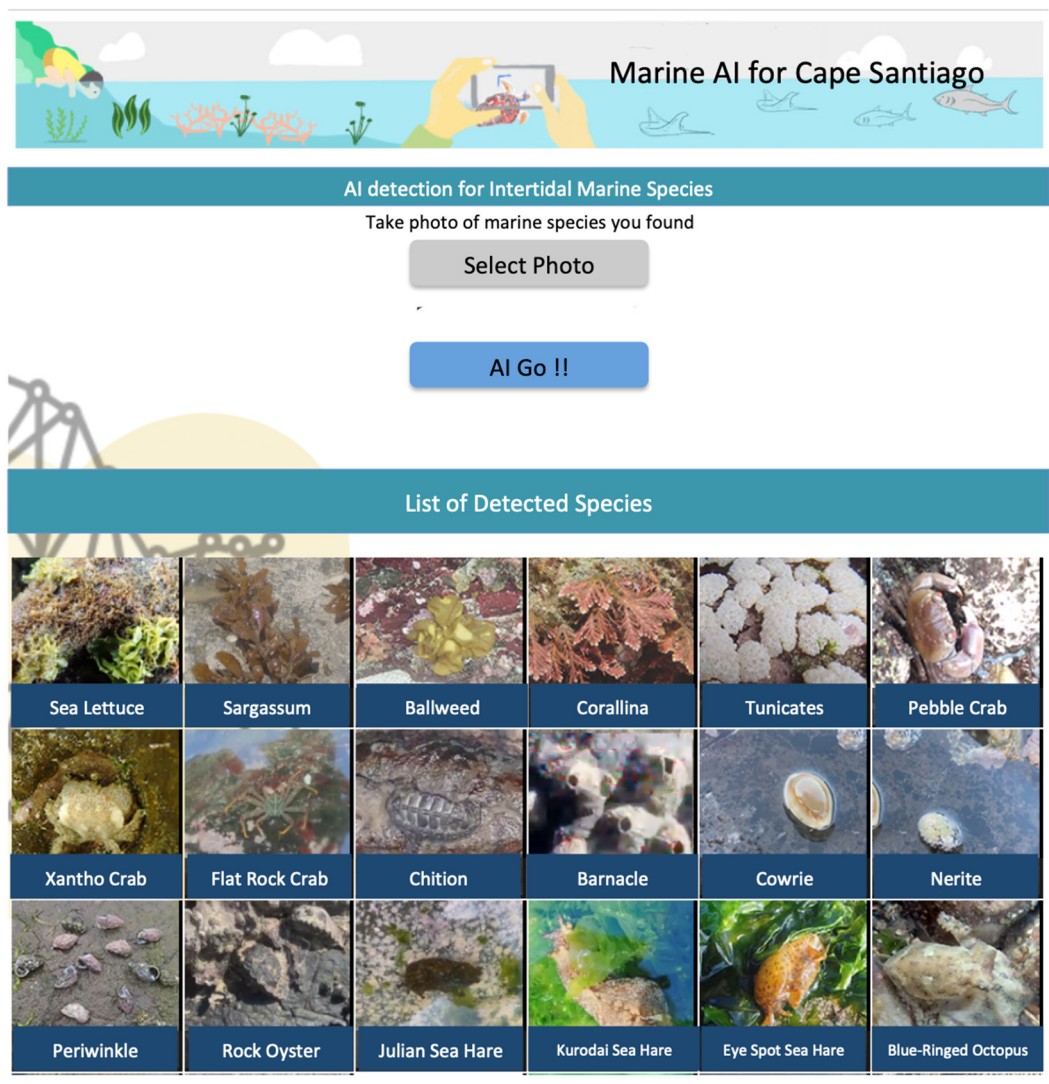

**Figure 6.** The program interface API incorporates an AI recognition service. Users can utilize the provided AI recognition capabilities to search for organism in the field. When they encounter a suspected specimen (pictures listed below as a guide), they can capture a photo and upload it from their mobile phones to the server to request AI identification.

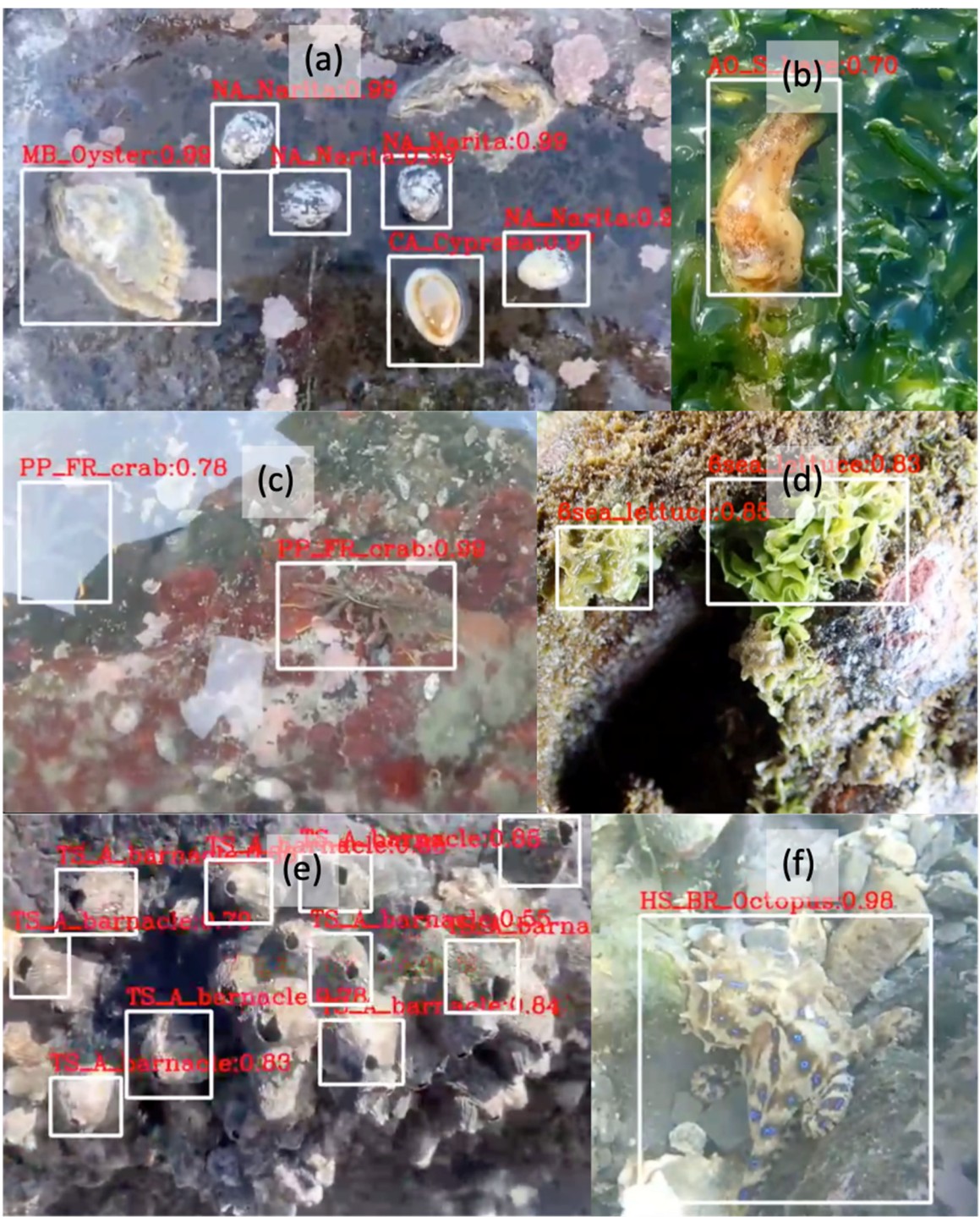

**Figure 7.** Examples of AI-identified specimen instances: (**a**) 4 blotched-nrites, 1 cowrie, and 1 rocky oyster, (**b**) 1 eye spot sea hare, (**c**) 2 flat rock crabs, (**d**) 2 sea lettuce, (**e**) 10 barnacle shells, (**f**) 1 blue-ringed octopus.

## 4. Discussions

### 4.1. Adaptive Survey Protocol for Rocky Intertidal Biodiversity

The Ocean Conservation Administration's proactive survey [48] initiative aligns with the prescriptive Marine Conservation Act draft, aiming to foster a deeper understanding of Taiwan's extensive coastline, spanning approximately 1600 km, including the rocky intertidal ecosystems. Among the 67 identified survey sites, each was meticulously assessed

for environmental and biological diversity. This assessment entailed a 60 m transect-line study from the high tide to the low tide mark, where researchers diligently documented observable marine species within a 60 min timeframe. Complementary to this, quadrat photographs were used to further analyze species presence and abundance.

However, the photographic method within the quadrat has its limitations, primarily disturbing mobile species, which may lead to an underrepresentation of non-sessile organisms. This method also struggles with capturing seasonal biodiversity variations, as demonstrated with the surveys conducted in early July at Cape Santiago. Seasonal dynamics, such as the absence of green algae due to summer conditions, could significantly skew the diversity index, omitting critical faunal dependencies.

The geological complexity of Cape Santiago's intertidal zones with its pits, grooves, crevices, and rock pools inherently impacts marine biodiversity. The intricate structures provide a multitude of microhabitats that support a rich array of life [58]. Davidson et al. [59] found that the diversity in mobile taxa can be twice as high as that in sessile taxa with sessile organisms presenting far greater abundance. This indicates that Cape Santiago's biodiversity is likely to exhibit considerable spatial and temporal variations that are influenced by monsoons, tides, water temperature, and sunlight exposure.

Addressing the spatial–temporal complexities and survey feedback from the CS participants, the adaptive survey protocol (ASP) was designed to overcome the limitations of the OCA transect-line method. It aims to prevent trampling damage to microhabitats by modifying survey routes to a single track along bench crevices (Figure 3), allowing for the CS participants to survey at their convenience, including nighttime. This flexible approach expands the spatial–temporal scope of data collection, which is essential for monitoring biodiversity changes.

### 4.2. Synergic Learning between CS and AI

In this study, a collaborative learning system that integrates CS and AI has been established. This system enables a symbiotic learning relationship where AI benefits from human-generated data and humans refine their knowledge through AI feedback, promoting mutual learning over extended periods. The flow of information and learning is illustrated in Figure 8. The CS participants, through workshops focusing on specimen identification and AI training, improve their skills and are, thus, able to contribute more effectively. They collectively gather extensive image datasets, 1301 images from PSP and 5461 images from ASP (shown in Figure 8c), which are crucial for effective PSP and ASP YOLOv5s model training. The outcomes of the training then provide insights, 0.96175 precision for PSP and 0.87412 precision for ASP, into the collective data quality of different survey protocols (Figure 8d).

Utilizing an online AI tool, the CS participants enhance the AI's identification capabilities through real-time data provision. This iterative process may refine the YOLOv5s model, which is our offline solution algorithm (offSA) that, after rigorous training using images from CS activities, is now operational on a server (Figure 8e). A user-friendly interface (Figure 6) has been established to streamline access to the online solution algorithm (onSA), serving both the CS participants (CSPs) and tourists engaged in educational programs (Figure 8a). The interface offers instant identification results (Figure 8b) and is integral to data collection for continuous AI training. This strategic implementation forms a dynamic feedback loop with data from queries contributing to successive training cycles, progressively enhancing the AI model's accuracy (Figure 8a,f).

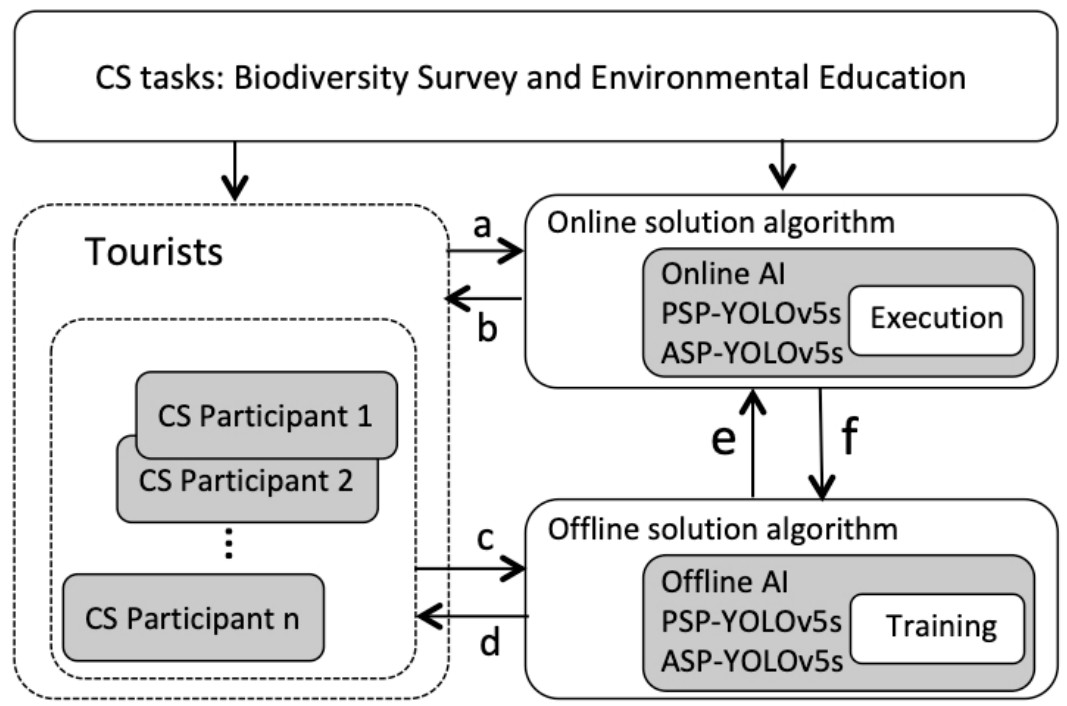

**Figure 8.** The information flow diagram (IFD) illustrating "Tier 3: hybrid intelligence (HI) at the collective level" adapted from Ref. [43]. Please refer to the accompanying text for details.

A crucial aspect of citizen science (CS) projects is maintaining data quality, a task complicated by the varied experience levels of the contributors [60–62]. This study shifts from individual to collective intelligence evaluations using AI model performance as a proxy for data quality assessment. Participants engaged in PSP may question whether their dataset of 1301 images with 2643 annotations sufficiently trains the PSP YOLOv5s model. The AI's performance, precision of 0.96175, recall of 0.93533, and mean average precision (mAP_0.5) of 0.95485, suggests robust training efficacy. Similarly, those conducting ASP with 5461 images and 7729 annotations must consider if their dataset is adequate. The ASP model's results, precision of 0.87412, recall of 0.85843, and mAP_0.5 of 0.8707, pose questions about disparities in performance. An inquiry arises: What factors contribute to these differences, and can the ASP model achieve the PSP model's level of accuracy?

Through confusion matrix analysis, the research evaluates the dataset quality for different models. Table 2, derived from the confusion matrices (Figures 4 and 5), details instances and their corresponding true positives (TP) recognized by the models. Certain species, such as the red coral algae (*Corallina* sp.) and tunicates, exhibit TP values that do not meet the desired standard. Despite their substantial representation in the training dataset with 301 and 512 instances, respectively, they achieved TP values of only 0.56 and 0.57. This stands in stark contrast to the Kurodai Sea Hare that, with merely 112 instances in the ASP dataset, secured a TP value of 0.98 (as shown in Table 2.). This discrepancy signals a challenge for the AI model in recognizing these specimens accurately, highlighting a need for improved data quality in both collection and annotation by the CS participants. Thus, the CS participants learn that data collection and annotation processes require further enhancement for better AI training outcomes.

**Table 2.** The instance amount and true positive (TP) values of different species data collected with the proposal survey protocol (PSP) and adaptive survey protocol (ASP).

| Common Name | PSP | | ASP | |
|---|---|---|---|---|
| | Instance | Species TP | Instance | Species TP |
| Eye Spot Sea Hare | 499 | 0.97 | 489 | 0.97 |
| Julian Sea Hare | 210 | 0.98 | 210 | 0.98 |
| Kurodai Sea Hare | 112 | 1.00 | 112 | 0.98 |
| Barnacle | 812 | 0.83 | 804 | 0.84 |
| Rock Oyster | 205 | 0.87 | 205 | 0.90 |
| Cowrie | 276 | 1.00 | 276 | 0.96 |
| Blotched Nerite | 312 | 0.99 | 312 | 0.99 |
| Blue-Ringed Octopus | 145 | 1.00 | 145 | 0.94 |
| Flat Rock Crab | 72 | 1.00 | 72 | 1.00 |
| Goby | | NA | 632 | 0.71 |
| Chition | | NA | 195 | 0.93 |
| Potamidids | | NA | 1266 | 0.87 |
| Sargassum | | NA | 781 | 0.79 |
| Sea Roach | | NA | 478 | 0.75 |
| Crescent Grunter | | NA | 298 | 0.77 |
| Sea Lettuce | | NA | 640 | 0.85 |
| Swimming Crab | | NA | 275 | 0.96 |
| Pebble Crab | | NA | 202 | 1.00 |
| Xantho Crab | | NA | 192 | 0.98 |
| Red Coral Algae (Corallina) | | NA | 301 | 0.56 |
| Ballweed | | NA | 280 | 0.96 |
| Sponge | | NA | 267 | 0.89 |
| Tunicates | | NA | 512 | 0.57 |
| Pyramid Periwinkle | | NA | 737 | 0.97 |
| Brittle Star | | NA | 213 | 1.00 |
| Damsel Fish | | NA | 221 | 0.78 |

### 4.3. The Criteria for Developing Hybrid Intelligence

Ref. [57] suggests three criteria for the development of a hybrid intelligence system. The first criterion, "collectiveness" within the context of HI pertains to the concerted effort of CS and AI to fulfill a system-level objective. In this study, the YOLO object-detection model necessitates an extensive array of specimen photographs with annotated instances for effective model training, a process reliant on the gradual and cumulative data collection by various CS participants over multiple surveys. Since individual contributions may be limited due to time constraints and unpredictable specimen appearances, aggregating data across different times is crucial for capturing species diversity. AI, thus, emerges as a catalyst, unifying the efforts of the CS participants toward a common goal, the comprehensive documentation of rocky intertidal ecosystem biodiversity.

The second criterion for effective HI is the achievement of "superior outcomes" compared to those possible with CS or AI in isolation. Repetitive surveys and environmental education tasks are often taxing, potentially diminishing volunteer enthusiasm. However, the assurance that their diligent survey efforts contribute to AI databases for educational purposes coupled with the positive feedback received from leveraging AI's novel technology sustain participant motivation and ensure ongoing involvement, a testament to the project's operational success.

Lastly, the establishment of "perpetual learning" constitutes the third criterion of HI. The AI's confusion matrix serves as a tool for assessing the quality of data for specific species, encouraging CS participants to persist in their data collection efforts. This ongoing collection process is not just during formal surveys but also through interactive specimen identification using AI during educational activities. As data accumulate, they refine the training dataset, enhancing AI's recognition capabilities. Moreover, AI becomes an

invaluable aid for newcomers in CS, facilitating rapid species identification skill acquisition and perpetuating the educational growth of the entire CS community.

### 4.4. AI and Citizen Science: A Socio-Technological Synthesis

Our study underscores the importance of not only building an AI model but also managing the socio-technological integration between AI and CS to ensure the effective functioning of HI. Inspired by McClure et al. [45], who delineated eight key attributes critical to CS and AI integration, we delved into these attributes, as they provide a comprehensive framework for managing HI systems within our research domain. These attributes and their complex interrelationships with AI and CS, alongside their associated tasks, are depicted in a mind map (Figure 9) and are further dissected in the subsequent sections.

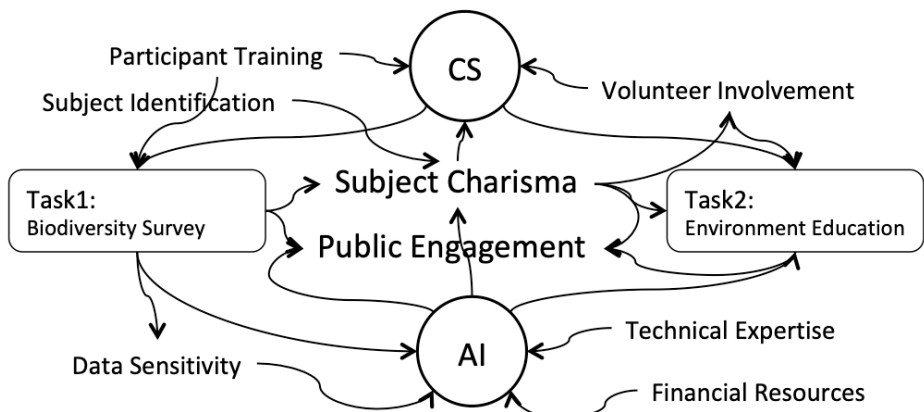

**Figure 9.** The illustration depicts the integrative mind map of the hybrid intelligence system, encompassing components of CS and AI, along with two distinct tasks. The map is further characterized by eight attributes crucial for effective system management. The directional arrows within the diagram indicate the influences between components and attributes. Detailed explanations can be found in the accompanying text.

11. Subject Charisma:

The intertidal zone of reef rocks teems with diverse marine life and is subject to dynamic ecological changes, presenting urbanites from terrestrial settings with a vivid and elusive tableau of the sea's inhabitants. These marine creatures, often sporting distinctive and vibrant appearances, make the zone an accessible, enigmatic, and captivating new realm that beckons for repeated exploration. Notably, charismatic species like the Eye Spot Sea Hare (*Aplysia oculifera*), which bears a resemblance to the popular cartoon character Pikachu, play a crucial role in heightening the involvement of CS participants and visitors. The combination of such appealing creatures and the deployment of smartphone-based AI tools for specimen identification enhances the marine biodiversity exploration into an engaging and educational ecological quest. This synergy significantly boosts environmental education initiatives and fosters public engagement.

12. Subject Identification:

Specimen recognition represents a considerable challenge, particularly for tourists with limited knowledge of marine biodiversity. Our study's protocol addresses this by providing a list of species that accounts for their varied appearances throughout the seasons. Such measures ensure that tourists can identify marine life during their visit, enriching their encounter with marine biodiversity and making the identification process an educational experience in its own right.

13. Public Engagement:

The hybrid intelligence-driven online AI service introduced in this study is a testament to the power of engaging the public in environmental stewardship. By enabling visitors to

identify organisms within the intertidal zone through an engaging process, we not only educate but also discourage detrimental behaviors, such as the capture of marine life. This strategy is key to garnering support for the conservation efforts outlined in 8. financial considerations.

14.   Volunteer Involvement:

The success of CS initiatives is heavily reliant on the dedication of volunteers. Despite the logistical challenges posed by the remote location of our study site, the dynamic nature of marine biodiversity here continually inspires local volunteers. The incorporation of AI technology into their routine surveys has been instrumental in sustaining their enthusiasm, as they often discover previously unrecorded species, underscoring the value of their involvement and the importance of their continued participation.

15.   Participant Training:

Training for CS participants is designed to be both accessible and practical, utilizing localized survey protocols and tailored species lists to navigate the complexities of marine biodiversity. Ongoing training workshops are scheduled to mitigate emerging challenges and ensure participants are well-equipped to conduct AI-assisted field surveys, which are vital for the AI's continuous learning, as highlighted in 6. technical expertise.

16.   Technical Expertise:

The gradual elevation of expertise among participants through hands-on surveys and engagement with online resources or marine experts highlights the mutual learning process central to HI. By contributing to image annotation, participants provide valuable data that enhance the AI training process and the model's subsequent performance.

17.   Data Sensitivity:

Sensitive handling of the images uploaded by participants for AI identification is paramount. Our protocol ensures that users are aware their contributions are confidential and used solely for the purpose of enhancing the AI model. Such transparency is essential in maintaining trust and encouraging the responsible sharing of data.

18.   Financial Considerations:

The choice between establishing a dedicated AI computing infrastructure versus using cloud AI services involves significant financial deliberations. Our decision to develop our own system reflects our commitment to research flexibility. Yet, the financial sustainability of both the AI infrastructure and the CS project remains a key concern, one that is necessary for the unbroken operation of HI and the achievement of our conservation goals.

By dissecting and addressing these attributes, our research not only provides a technical blueprint for AI–CS integration but also offers insights into the socio-technological considerations that underpin the successful deployment of HI systems for environmental conservation.

### 4.5. The Role of Hybrid Intelligence in MPA Designation

Designing a marine protected area (MPA) is a complex task. As per the recent updates to the US National Marine Sanctuaries Act, effective management and a comprehensive understanding of the area over a decade are prerequisites [63]. Addressing these challenges, particularly those related to personnel resources, the OCA report has indicated the valuable role citizen scientists could play [48].

By collating specimen survey data to create training materials, CS participants can perform year-round, consistent monitoring within specified areas. This method paves the way for a deeper understanding of the seasonal patterns within marine communities and yields more detailed data on the temporal and spatial diversity of marine ecosystems.

HI, which merges the collective efforts of citizen science and artificial intelligence (AI), establishes a progressive framework for marine species education and motivates the public to partake in regular spatiotemporal surveys. This process of collecting data at different

times and locations is crucial to meeting the comprehensive training and data collection demands essential for setting up an MPA.

Beyond the ecological considerations, MPAs situated near urban areas face a wider spectrum of socio-ecological challenges. Not only must they address the transformation of the economic activities of fishing communities, but MPAs in suburban locales also present opportunities for urbanites to reconnect with nature, alleviating the stress of confined urban living conditions [64]. Through participatory design involving the community and the public, HI serves as a biophilic instrument that enhances urban livability.

## 5. Conclusions

Effective system management enhances the integration of CS and AI within the hybrid intelligence system, providing robust solutions for intertidal biodiversity conservation initiatives at Cape Santiago. Field surveys undertaken by trained CS participants generate image datasets crucial for AI training. The training outcomes, representing a transformation of multifaceted image data into numerical values, facilitate intertidal biodiversity analysis and reflect the overarching quality of data procured by all CS participants. Once an AI model is trained and available on the Web, it functions as a marine organism detection tool, supporting CS participants in their environmental education endeavors, especially for tourist instruction. Continuous feedback from user interactions precipitates holistic improvements in the HI system, encompassing the fine-tuning of survey methodologies, the prioritization of specific species, and the customization of AI tools to meet the demands of in situ environmental education. Strategically managing this HI not only fosters sustained engagement among CS participants but also bolsters broader public involvement, thereby strengthening advocacy for the prospective designation of the region as a marine protected area.

**Author Contributions:** Conceptualization, V.Y.C., D.-J.L. and Y.-S.H.; Data curation, V.Y.C.; Formal analysis, V.Y.C.; Funding acquisition, Y.-S.H.; Investigation, V.Y.C.; Methodology, V.Y.C.; Resources, Y.-S.H.; Software, V.Y.C.; Supervision, Y.-S.H.; Validation, V.Y.C. and D.-J.L.; Writing—original draft, V.Y.C.; Writing—review and editing, V.Y.C. and D.-J.L. All authors have read and agreed to the published version of the manuscript.

**Funding:** The National Science and Technology Council, Executive Yuan, Taiwan (MOST 111-2313-B-002-016-MY3).

**Institutional Review Board Statement:** Not applicable.

**Informed Consent Statement:** Not applicable.

**Data Availability Statement:** Data are contained within the article.

**Acknowledgments:** We appreciate the support of the CHENG CHEN foundation, SDCDA, and the invaluable contributions of all participating volunteers in gathering AI training data and providing feedback for this study. Additionally, our sincere thanks go to the anonymous reviewers, whose insightful comments significantly enhanced the quality of this manuscript.

**Conflicts of Interest:** The authors declare that they have no known competing financial interests or personal relationships that could have appeared to influence the work reported in this paper.

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
