# Peer review of "Hybrid Intelligence for Marine Biodiversity: Integrating Citizen Science with AI for Enhanced Intertidal Conservation Efforts at Cape Santiago, Taiwan"

_sustainability, doi:10.3390/su16010454_

Round 1
Reviewer 1 Report
Comments and Suggestions for Authors
The article explores the integration of Artificial Intelligence (AI) in the assessment of biodiversity within intertidal coastal zones, offering an insightful perspective into the innovative advancements shaping environmental conservation. By harnessing AI algorithms and machine learning techniques, the research exemplifies a promising avenue for the rapid and accurate evaluation of diverse marine ecosystems. However, I do have some minor concerns:
Introduction is too long, it should be shortened, especially the text of page 2 regarding human activities and impacts.
My main concern is the Methodology, Results and Discussion Section, in my opinion they should be rewritten, considering that part of the methods is in the results and part of the results in the discussion. The protocols and AI training specifics should be in the methodology for example.
Also the figures are of poor quality, so they should be altered.
I think that a carefull review of english should be enough as mostly is well writen but sometimes sentences does not make sense, for example lines 171-172.
Comments on the Quality of English LanguageI think that a carefull review of english should be enough as mostly is well writen but sometimes sentences does not make sense, for example lines 171-172.
Author Response
First and foremost, we would like to thank you for their constructive feedback and valuable suggestions. We have thoroughly revised the manuscript in light of your comments and have made several changes to enhance its quality. Below, we provide a point-by-point response to each comment:
[Comment 1]: Introduction is too long, it should be shortened, especially the text of page 2 regarding human activities and impacts.
[Response 1-1]: Manuscript line 41-71 two paragraphs has been shortened into one paragraph as following:
The Anthropocene epoch, marked by significant human impact on Earth's biophysical systems, has led to intensified pressures on coastal regions due to population growth, urbanization, and climate change [16]. This has notably affected rocky intertidal ecosystems, with human interaction evolving from fisheries exploitation to recreational activities like adventure tourism [17]. A multifaceted coastal management approach, incorporating diverse strategies from legal to educational, is essential to mitigate environmental impacts and encourage societal participation [18]. Additionally, the largely unexplored marine domain presents opportunities for Marine Citizen Science (MCS). MCS plays a crucial role in filling research gaps and enhancing global marine conservation efforts [19, 20]. It provides cost-effective, robust data that informs policy decisions [21, 22] while increasing public science literacy and community engagement in marine issues [23, 24, 25], from cetacean conservation [26, 27] to addressing plastic pollution [28], thereby enabling community-driven initiatives to transform research into effective policies [29, 30].
[Response 1-2]: And, manuscript l ine 97-123 two paragraphs has been shortened into one paragraph:
Cape Santiago, located on Taiwan Island's easternmost point, near Taipei, is at the intersection of the Kuroshio and longshore currents. Named by Spanish explorers 400 years ago, its 4.86km coastline features wave-cut benches and rocky shores, and is home to the small Magang fishery harbor community. With declining fishery resources driving youth to cities, the local government, aiming to boost tourism, built a bicycle route to the coast in 2011. However, the influx of tourists engaging in harmful activities like capturing marine species has raised concerns about the impact on biodiversity and the coastal environment. The Cape Santiago Culture Development Association (SDCDA) was formed to protect the fishery village culture and marine ecosystem. Legally, Cape Santiago's coast is an "Ocean Resource Protected Area" under the Urban Planning Law [46], restricting construction but not specifically addressing environmental harm.
[Comment 2]: Rewrite the specifics of protocols and AI training.
[Response 2-1]: The survey protocol has been rewritten as following
2.1. Study area and the Citizen Scientist Project
The survey protocol, derived from the Ocean Consecration Administration’s (OCA) transect line method, required two individuals to methodically search and document marine species along a designated transect line, spanning from the high tide zone to the low tide area of the shore. This procedure was conducted by volunteers who had previously received Citizen Science (CS) workshop training. During the survey, participants followed a pre-set route marked by the transect line, photographing all marine species encountered and recording the quantities of specific target species on survey forms. Following the survey, the collected data and images were systematically compiled and uploaded to create a comprehensive training dataset for future artificial intelligence (AI) training sessions.
We also modify the correspond results:
3.1. Survey Protocols
“….Survey groups, each comprising 2 members, aligned with these lines. Therefore, a total of 10 groups worked in tandem, collecting data and photographs for AI training…….Upon implementation, some challenges with the original Proposal Survey Protocol (PSP) (Figure 3 (a)) became apparent. Some species, like the sea hare, were absent during the summer surveys. Considering the route's dual function as a prospective educational pathway, a looped configuration was determined to be more suitable than a linear design. This feedback led to the development of an Adaptive Survey Protocol (ASP) (Figure 3 b)…..”
Finally, we discuss the reasons and methods behind the improvements to the survey protocol in Section 4.1, which follows the logical structure of the protocol topics.
Please refer “4.1. Adaptive survey protocol for rocky intertidal biodiversity”
[Response 2-2]: AI training parts
Section 2.2 on AI training has been expanded to include more information, thereby clarifying the training methods used:
2.2. Training AI Model and AI Performance Evaluations
…. The images of organisms from the participants' field survey dataset were meticulously identified and annotated as 'instances' using the open-source Python program labelImage (https://github.com/HumanSignal/labelImg). Each instance was marked as a bounding box with its location, size, and common name within the image. Effective AI recognition of a particular species necessitates a substantial image database for that species, typically comprising 200 or more instances, as per our experience. Consequently, species that are commonly observed at the research site are usually selected for AI training. To keep participants engaged, we also include a selection of rare target species, like sea hares and blue-ringed octopuses, encouraging participants to search for and document these less frequently encountered organisms.……..
The results of AI training were explained at section 3.2.
“…The total annotated instances across all species amounted to 2,643 (Figure 4 (a)). Utilizing this dataset, we trained the PSP YOLOv5s neural network, achieving a performance marked by precision: 0.96175, recall: 0.93533, and mAP_0.5: 0.95485. An examination of the confusion matrix indicated high true positive (TP) values for almost all species. Notably, barnacles and oysters registered TP values of 0.83 and 0.87, respectively. All other species achieved impressive TP values exceeding 0.97, with some even reaching a perfect score of 1.0 (100%) (Figure 4 b).”
“….Leveraging this enriched dataset, we trained the ASP YOLOv5s model, which yielded a performance evaluation of precision: 0.87412, recall: 0.85843, and mAP_0.5: 0.8707. A deeper dive into the confusion matrix, post enhancement of instance and species counts, revealed that the True Positive (TP) values for species like red coral algae (Corallina sp.) and tunicates were the lowest, registering at 0.56 and 0.57, respectively……”
At last, we discuss the how CS make AI training/learning follows the logical structure of the AI and CS mutual learning topics.
4.2. Synergic Learning between CS and AI
“…..The outcomes of the training then provide insights, 0.96175 precision for PSP and 0.87412 precision for ASP, into the collective data quality of different survey protocol (Figure 8d)…”
[Comment 3]: poor quality of figures and s
[Response 3]: The quality of the figures has been enhanced with high-resolution images, and Figure 4 has been redrawn with increased font size for the numerical data.
[Comment 4]: lines 171-172 sentences does not make sense
[Response 4]: This sentence has been revised to “The survey protocol, derived from the Ocean Consecration Administration’s (OCA) transect line method, required two individuals to methodically search and document marine species along a designated transect line, spanning from the high tide zone to the low tide area of the shore”, making it clearer and more understandable.
Please refer to the attached revised manuscript; the updated text is highlighted in red, and the revised figures are also included for your review.
We believe these modifications have significantly enhanced the quality and clarity of our work. Once again, we would like to express our sincere gratitude to you for the time and effort spent reviewing our manuscript.
Reviewer 2 Report
Comments and Suggestions for Authors
Thank you for giving me this opportunity to review the manuscript entitled “Hybrid intelligence for marine biodiversity: Integrating citizen science with AI for enhanced intertidal conservation efforts at Cape Santiago, Taiwan.”
This study is well designed and used an innovative approach. This study can contribute to using AI for preservation of MPAs. I would like to suggest this manuscript for publication, however, I have some comments.
1. Table 1
The locations of the research sites in Table 1 can be all checked in Figure 1. Lai-Lai 1, 2, and 3 are not clear for me who are not familiar with the sites.
2. species, the subjects of the research
I believe there would be more and various species in the research sites. What are the criteria for the training? Are the species in the collected data in the list of Table 2 or what are they, examples of species?
3. Figure
What is instances in Figure 4(a) and Figure 5(a)?
“1,301 images from PSP and 5,461 images from ASP 369 (shown in Figure 7c)” should be more clearly noted in the results. Adding the total of the images can be helpful in the tables.
4 .Figure
The figures 4(b) and Figure 5(b) are too small to read the information.
5. figure
The data of figure 6 and figure 7 seem to be taken on the surface or sallow sea side. So, this study did not cover diverse species of MPAs. More specific information regarding the data should be noted.
Author Response
First and foremost, we would like to thank you for their constructive feedback and valuable suggestions. We have thoroughly revised the manuscript in light of your comments and have made several changes to enhance its quality. Below, we provide a point-by-point response to each comment:
[Comment 1]: The locations of the research sites in Table 1 can be all checked in Figure 1.
[Response 1]: The location of Lai-Lai 1, 2, and 3 have been marked at Figure 1.
[Comment 2]: What are the criteria for the AI training
[Response 2]: Effective AI recognition of a particular species necessitates a substantial image database for that species, typically comprising 200 or more instances, as per our experience. Consequently, species that are commonly observed at the research site are usually selected for AI training. To keep participants engaged, we also include a selection of rare target species, like sea hares and blue-ringed octopuses, encouraging participants to search for and document these less frequently encountered organisms.
[Comment 3]: What is instances in Figure 4(a) and Figure 5(a)?
[Response 3]: An instance refers to an image of a species highlighted with a bounding box, indicating its location, size, and common name within the photo. For further details, please refer to the revised method section 2.2.
[Comment 4]: The figures 4(b) and Figure 5(b) are too small to read the information.
[Response 4]: The quality of the figures 4(b) and 5(b) has been enhanced with high-resolution images to enhance clarity, and figure 4(b) has also been redrawn with increased font size for the numerical data.
[Comment 5]: The data of figure 6 and figure 7 seem to be taken on the surface or sallow sea side. So, this study did not cover diverse species of MPAs?
[Response 5]: Figure 6 shows the Application Program Interface (API) screen used by users to access AI recognition services, while Figure 7 presents an example of the identification results obtained by users after using AI for species recognition.
Please refer to the attached revised manuscript; the updated text is highlighted in red, and the revised figures are also included for your review.
We believe these modifications have significantly enhanced the quality and clarity of our work. Once again, we would like to express our sincere gratitude to you for the time and effort spent reviewing our manuscript.
Reviewer 3 Report
Comments and Suggestions for Authors
Comments to authors on file

Author Response
First and foremost, we would like to thank you for their constructive feedback and valuable suggestions. We have thoroughly revised the manuscript in light of your comments and have made several changes to enhance its quality. Below, we provide a point-by-point response to each comment:
[Comment 1]: should shorten the Introduction section by removing the description and map of the study area and move these blocks to the Materials and Methods section.
[Response 1-1]: Manuscript line 41-71 two paragraphs has been shortened into one paragraph as following:
The Anthropocene epoch, marked by significant human impact on Earth's biophysical systems, has led to intensified pressures on coastal regions due to population growth, urbanization, and climate change [16]. This has notably affected rocky intertidal ecosystems, with human interaction evolving from fisheries exploitation to recreational activities like adventure tourism [17]. A multifaceted coastal management approach, incorporating diverse strategies from legal to educational, is essential to mitigate environmental impacts and encourage societal participation [18]. Additionally, the largely unexplored marine domain presents opportunities for Marine Citizen Science (MCS). MCS plays a crucial role in filling research gaps and enhancing global marine conservation efforts [19, 20]. It provides cost-effective, robust data that informs policy decisions [21, 22] while increasing public science literacy and community engagement in marine issues [23, 24, 25], from cetacean conservation [26, 27] to addressing plastic pollution [28], thereby enabling community-driven initiatives to transform research into effective policies [29, 30].
[Response 1-2]: Additional, manuscript line 97-123 two paragraphs has been shortened into one paragraph:
Cape Santiago, located on Taiwan Island's easternmost point, near Taipei, is at the intersection of the Kuroshio and longshore currents. Named by Spanish explorers 400 years ago, its 4.86km coastline features wave-cut benches and rocky shores, and is home to the small Magang fishery harbor community. With declining fishery resources driving youth to cities, the local government, aiming to boost tourism, built a bicycle route to the coast in 2011. However, the influx of tourists engaging in harmful activities like capturing marine species has raised concerns about the impact on biodiversity and the coastal environment. The Cape Santiago Culture Development Association (SDCDA) was formed to protect the fishery village culture and marine ecosystem. Legally, Cape Santiago's coast is an "Ocean Resource Protected Area" under the Urban Planning Law [46], restricting construction but not specifically addressing environmental harm.
[Response 1-3]: Map of the study area and introductory text blocks have been moved to methods section.
[Comment 2]: removing unnecessary non-scientific descriptions (lines 113-114). Lines 135-136 are also redundant in a research paper.
[Response 2]: Thank you for suggestion, all these text blocks have been rewritten and moved to methods section.
2.1. Study area and the Citizen Scientist Project
The study area of this paper is located on the northwest side of the Cape Santiago coastline, specifically focusing on a wave-cut bench, termed the Santiago bench. This bench spans approximately 160 meters in length and 70 meters in width (Figure 1). The primary aim of the survey is to identify and document target marine organisms within this predefined area.
To evaluate the potential Marine Protected Areas (MPAs), the Ocean Consecration Administration (OCA) conducted biodiversity assessments at 67 rocky shores spanning Taiwan's main island and its surrounding islands [48]. They utilized transect line and quadrat methodologies to assess marine species richness, biomass, and a range of environmental factors including the impacts of sewage outfalls, man-made facilities, tourist activity, siltation levels, and the abundance of rock pools. By integrating both biodiversity and environmental indicators, the OCA determined the conservation priorities of these 67 rocky shores. Among the surveyed locations were four sites at Cape Santiago, including Santiago bench, Lai-Lai-1, Lai-Lai-2, and Lai-Lai-3. Notably, the Lai-Lai-2 site, located at the southeastern edge of Cape Santiago, showcased the greatest marine species diversity. Along with the neighboring Lai-Lai-3 site, both were identified as having the highest conservation value, as reported in OCA's pilot survey [48].
[Comment 3]: 4 sites should be indicated
[Response 3]: The location of Lai-Lai 1, 2, and 3 have been marked at Figure 1.
[Comment 4]: Legend of Table 2 is not proper, and error in the word "comon" at Table 3.
[Response 4]: Table legend has be modified as “The structure of the macrobenthos in the study area of marine species of Santiago bench.”, and the error spelling of “comon” has been corrected.
[Comment 5]: Resolution and font size of figure 4 (b) and deciphering of all abbreviations
[Response 5]: The quality of the figures has been enhanced with high-resolution images, and Figure 4 has been redrawn with increased font size for the numerical data. Deciphering of all abbreviations: high water level (HTL), low water level (LTL), winter (W), spring (SP), summer (SU), autumn (F) at both captions of Figure 4 and Figure 5.
[Comment 6]: Figure 6 should be translated into English
[Response 6]: The characters of Figure 6 has been translated into English.
[Comment 7]: “species identification” of Figure 7
[Response 7]: Yes, the use of the word 'species' is not appropriate; it should be corrected to 'specimen'.
[Comment 8]: Redundancy of figure 7 at section 4.2. and question of ‘species are identified by whom - AI? Or is it a joint identification by AI + human?’
[Response 8]: In Section 4.2., the references to 'Figure 7a, 7b...' were incorrect and have been updated to 'Figure 8a, 8b...'. These figures accurately illustrate the combined identification process involving both AI and human input.
[Comment 9]: Question of ‘Why are only 10 barnacles be identified? Not all of them?’.
[Response 9]: The example in Figure 7(e) demonstrates the limitations in detection capability of our AI model. It cannot fully detect or identify every specimen in an image, indicating a need for more data to further train and enhance its performance.
[Comment 10]: The issue of living or non-living organism identification.
[Response 10]: The example in Figure 7(e) illustrates the AI model identifying barnacles, which may include dead specimens. These should be noted as 'barnacle shells.' It's important to mention that our AI model cannot distinguish between living and non-living barnacles, as this distinction was not incorporated into the training images used to develop the AI model.
[Comment 11]: Adding a similar figure for macrophyte identification at figure 7.
[Response 11]: The example image showing the successful identification of the macrophyte sea lettuce is provided in Figure 7(d) for your reference.
[Comment 12]:Question about “ species-level analysis” at “synergic Learning between CS and AI section”,
[Response 12]: We wholeheartedly concur with your comments regarding the 'species-level analysis.' Given that the entire paper focuses on intertidal biodiversity, we agree that changing this to 'intertidal biodiversity analysis' would be more appropriate and consistent. We would like to extend our gratitude once again for your insightful feedback.
Please refer to the attached revised manuscript; the updated text is highlighted in red, and the revised figures are also included for your review.
We believe these modifications have significantly enhanced the quality and clarity of our work. Once again, we would like to express our sincere gratitude to you for the time and effort spent reviewing our manuscript.
Round 2
Reviewer 3 Report
Comments and Suggestions for Authors
I thank the authors for their work in improving the manuscript.
I would like to ask the authors to check again the whole text for correctness of the use of terms, namely "species identification". It would be more appropriate to introduce another terminology "identification of organisms" or something else.
For example, but not limited to:
Line 303-304 Crafted with the intent to aid CS participants during surveys or environmental educational endeavors, this tool proves invaluable in real-time species identification.
Author Response
Once again, we greatly appreciate your valuable suggestions. Concerning the use of the term 'species,' we have thoroughly revised the manuscript and made several changes. We have replaced 'species' with either 'specimen' or 'organism,' depending on the context, as illustrated in the following two paragraphs and in several other sections, which are highlighted in blue in the second revised manuscript.
Revised 'species' at Line 303-313 paragraph and others at manuscript:
3.4. Deployment AI for In-Situ Identification An application program interface (API) was developed using the Python programming language to facilitate a web-based interface (Figure 6). This interface hosts the trained AI model dedicated to marine specimen recognition and is stationed on a cloud server for efficient specimen identification tasks. Crafted with the intent to aid CS participants during surveys or environmental educational endeavors, this tool proves in-valuable in realtime specimen identification. On encountering marine organisms in their natural habitat, users can snap photos using their mobile devices and promptly upload them for instant AI-based identification. Upon receiving the image, the API communicates with the AI model to carry out the recognition process, subsequently relaying the results back to the user (Figure 7). This streamlined process greatly enhances the efficiency and accuracy of specimen surveys and educational activities.
Figure 6. The program interface API incorporates an AI recognition service. Users can utilize the provided AI recognition capabilities to search for organism in the field. When they encounter a suspected specimen, which pictures listed below as a guide, they can capture a photo and upload it from their mobile phones to the server to request AI identification.